# Comparison of Energy Used and Effects on Bulk Density and Yield by Tillage Systems in a Semiarid Condition of Mexico

Alfredo López-Vázquez [1], Martin Cadena-Zapata [2,*], Santos Campos-Magaña [2], Alejandro Zermeño-Gonzalez [3] and Mario Mendez-Dorado [2]

[1] Production Systems Engineering Program, Universidad Autónoma Agraria Antonio Narro, Calzada Antonio Narro 1923 Saltillo, Coahuila 25315, Mexico; fred.max@live.com.mx

[2] Agricultural Machinery Department UAAAN Saltillo, Universidad Autonoma Agraria Antonio Narro, Saltillo, Coahuila 25315, Mexico; camposmsg@hotmail.com (S.C.-M.); mario.mendez@uaaan.edu.mx (M.M.-D.)

[3] Irrigation and Drainage Department UAAAN, Saltillo, Universidad Autonoma Agraria Antonio Narro, Saltillo, Coahuila 25315, Mexico; azermeno@uaaan.mx

* Correspondence: martin.cadena@uaaan.edu.mx; Tel.: +52-844-534-2349

**Abstract:** Energy used for tillage is an input with a high impact on the cost of crop production; it is desirable to till the soil using minimum energy. The objective of this study was to compare the specific energy expenditure, effect on soil bulk density, and forage yield of maize, by three tillage systems: Disk plow/Disk Harrow/planter (DDP), Chisel plow/Disk harrow/planter (CHDP) and No-Tillage (NT). Energy was measured for tillage operations in the summer season of 2013, 2016, and 2017. Bulk density in 2013 and 2016. Yield in 2013 and 2014. The variables of drawbar force (kN), working speed (m s$^{-1}$), width (m), depth (m), fuel consumption (L ha$^{-1}$), bulk density (g cm$^{-3}$), and dry matter yield (Mg ha$^{-1}$) were measured. Results showed that there were significant differences in the amount of energy used per ha; DDP used an average of 379.75 MJ, CHDP 135.01 MJ, and NT 26.43 MJ. The average energy applied to the soil mass for each system was 400 J kg$^{-1}$ for DDP, 255.13 J kg$^{-1}$ for CHDP, and for NT was 237.8 J kg$^{-1}$. The overall energy efficiency was; 18.23% for DDP, 6.88% for CHDP, and 4.77% for N. The bulk density decreased significantly after three years for NT. There were no significant differences in dry matter yield. In the semiarid condition of Mexico, CHDP and NT are options for saving from 64% to 93% of energy, compared with DDP.

**Keywords:** energy; tillage systems; fuel consumption; energy use efficiency; semiarid condition

## 1. Introduction

Soil tillage requires high energy input, normally diesel fuel [1–3]. In these times, it is necessary to make the most efficient use of all inputs, to have a sustainable crop production system [4].

Conventional tillage (DDP) implies a working depth from 0.25 m to 0.30 m for disk plowing with an intensive manipulation of a high volume of soil; thus requiring from five to nine times more energy compared with conservation tillage [5]. This intensive tillage can also lead to the degradation of soil structure, increasing the bulk density, and, thereby, affect porosity, hydraulic conductivity, and other soil properties that are important for the availability of water for crops [6].

Chisel Tillage (CHDP) avoids these drawbacks in two ways; it does not invert the soil, thus, preserving the soil structure, while leaving crop residues on the surface, which helps prevent water loss by evaporation. [7]. This type of conservation tillage decreases the intensity and frequency of soil disturbance, compared with conventional tillage [8]. No-Tillage (NT), by direct planting into

unprepared soil, causes the least disturbance to the soil. Direct seed planters cause a minimum of soil disturbance [9]. However, impacts of tillage systems on soil properties and yields are variable and are highly dependent on site conditions, such as soil type, climate, and production systems [10].

The cost of energy for tillage activities is increasing. International prices of diesel fuel rose 3.2 times in a period of nine years from 2004 to 2013 [11]. In Mexico, the prices of fuels derived from petroleum increased 2.3 times in the decade from 2007 to 2017 [12]. In field studies, it has been determined that up to 42% of energy for crop production is from the use of diesel [13].

With the continuous increase in prices for fossil fuels, producers look for options to diminish the amount of energy applied to food production systems [14]. The possibility of reducing production costs through the efficient use of energy is the main reason many farmers adopt conservation tillage systems, rather than a desire to preserve natural resources [15].

However, the use of energy should not be analyzed only from an economic point of view. It is important to consider that fossil fuels are being depleted and that their use is causing environmental problems [16]. Therefore, the use of tillage practices should be oriented not only to reducing energy expenditure but also to conserving natural resources [17]. The reduction of energy in tillage operations is one of the characteristics of conservation tillage systems. For example, [18] documented a decrease in the use of energy from 31% to 40% in conservation systems compared with a conventional production system of maize and wheat.

Management strategies, such as crop rotation and reduction of tillage practices, can improve efficiency in energy use [19]. Other studies show that from 10% to 30% of the applied energy for crop production could be saved by adopting best calibration and maintenance practices, as well as training operators [20].

Studies which simulate the variables that influence field fuel consumption for tractors have been conducted in an effort to find better use of energy. This allows growers to implement management strategies to maximize the performance and efficiency of field operations [21,22]. However, soil and climate conditions vary widely. Most studies of tillage methods have been conducted in temperate and humid climates. So, these results are not necessarily appropriate for the arid and semiarid climates [23,24].

The Northeast of Mexico is a semiarid region, cropland amounts 2.7 million ha. The main crops are forages (oats, alfalfa, maize) grain sorghum, grain maize, soybean, cotton, and pecan trees [25]. Conservation of water may be the most critical issue in the region. Soil erosion driven by wind and water is also a concern [26]. Policymakers and farmers need technical information to take appropriate soil management decisions to enhance soil quality and yields [27].

In some conditions, conservation tillage, (reduced tillage, vertical tillage, and no tillage), improves physical and biological soil properties [28]. Reducing tillage and maintaining crop residues on the soil surface can improve water use efficiency [29], aggregate stability, water infiltration and availability, and improve with conservation tillage systems, compared to conventional tillage [30]. In other cases, no tillage results in negative effects, increasing the bulk density in the surface, lowering the infiltration rates, and decreasing crop yield, compared with conventional tillage [31,32].

It is necessary to assess the performance under differing conditions, analyzing the efficiency of energy use and productivity, to have the criteria to determine the best methods and technologies [33].

The objective of this study was to compare energy usage, the effect on soil bulk density and crop yield among three tillage methods: DDP, CHDP, and NT. Specifically, we determined the energy per worked area (MJ ha$^{-1}$), the energy used for moving the soil mass (J kg$^{-1}$), the overall efficiency, the bulk density, and yield for the three methods. It was expected that in the semiarid condition of Mexico, the conservation tillage systems, CHDP and NT, would demand less fuel and energy, have a positive effect on bulk density and yield as compared with the conventional system (DDP), most commonly used in our area.

## 2. Materials and Methods

### 2.1. Study Site

Field experiments were conducted in the summer crop seasons of 2013, 2016, and 2017 at the experimental station of Universidad Autonoma Agraria Antonio Narro in Saltillo, Coahuila, Mexico. The site is located at 25°23′42″ N and 100°59′57″ W, at an altitude of 1743 m above sea level. The climate is semiarid with an average annual temperature of 16.9 °C, the mean annual rainfall 435 mm, and annual evaporation of 1956 mm.

The soil is a clay loam (34.1% clay, 33.4% silt, and 32.5% sand) with 2.09% of organic matter. Average soil bulk density for the site is 1.37 gr cm$^{-3}$. At each season just before the tillage operations, samples of soil were taken at each plot with an auger, weighed, and oven dry for 24 h at 105 °C and the moisture content calculated on weight basis. Average soil moisture content is presented in Table 1.

**Table 1.** Average soil moisture content in the soil profile (30 cm) at the experimental plots.

| Tillage Systems | Years | | |
|---|---|---|---|
| | **2013** | **2016** | **2017** |
| | Soil moisture content w/w | | |
| DDP | 11.20 | 11.73 | 5.32 |
| CHDP | 10.54 | 15.33 | 7.09 |
| NT | - | 8.89 | 6.62 |

Tillage systems: Disk plow/Disk Harrow/Planter (DDP), Chisel plow/Disk harrow/Planter (CHDP) and No-Tillage (NT).

### 2.2. Treatments and Experimental Setup

The three tillage treatments were applied in plots of 12 m × 40 m and replicated three times. A description of the characteristics of the implements is presented in Table 2.

**Table 2.** Characteristics of the implements used for the tillage systems.

| Characteristics | Disk Plow ARHK-3 | Chisel Plow JD 610 | Offset Disk Harrow RI 2024 | Planter JD Max Emerge |
|---|---|---|---|---|
| Working tools | 3 discs of 711 mm diameter and 6.35 mm thickness, Disk concavity 110 mm | 8 tines "C" type shank | Front section of 10 notched disks, one rear section of 10 plain disks with a disk spacing of 230 mm, diameter of disk of 609.6 mm and 4 mm thickness. Disk concavity 95 mm | 4 planting units at spacing of 910 mm, double disk opener |
| Weight of implement (kg) | 542 | 618 | 674 | 841 |
| Type of coupling to the tractor | Mounted, three-point linkage category 2 | | | |

The operations for the Disk Plow Tillage (DDP) were: Disk plowing (disk plow ARHK-3, Kimball, Torreon, COAH. Mexico), harrowing (disk harrow RI 20204, Tecnomec Agricola SA de CV, AGS. Mexico), and planting (planter JD Max Emerge 7000, John Deere SA de CV, Monterrey, NL, Mexico).

For Chisel Plow Tillage (CHDP), the operations were: Chisel plowing (Chisel plow JD610, John Deere SA de CV Monterrey, NL, Mexico), Disk harrowing, and planting.

For No-Tillage (NT), the only tillage operation was planting.

The collection of data in the field was made under a fully randomized experimental design. For the statistical analysis of the data we used R software. For the comparison of mean values, between treatments, we used the Tukey test with $\alpha \leq 0.05$.

### 2.3. Measurement of Variables in the Field

The following variables were measured during the soil preparation of the summer cropping season in 2013, 2016, and 2017.

The net force (NF) required for pulling the implements used in the tillage systems was measured using an integrated dynamometer for mounted tillage implements. The dynamometer consists of three octagonal extended ring (OER) transducers and a floating structure attached to the three-point linkage of the tractor [34]. The implements were coupled to the dynamometer as shown in Figure 1. The power source was a John Deere tractor model 6403, 2WD with a rated engine power of 73 kW. The front to rear weight distribution is 35% and 65%.

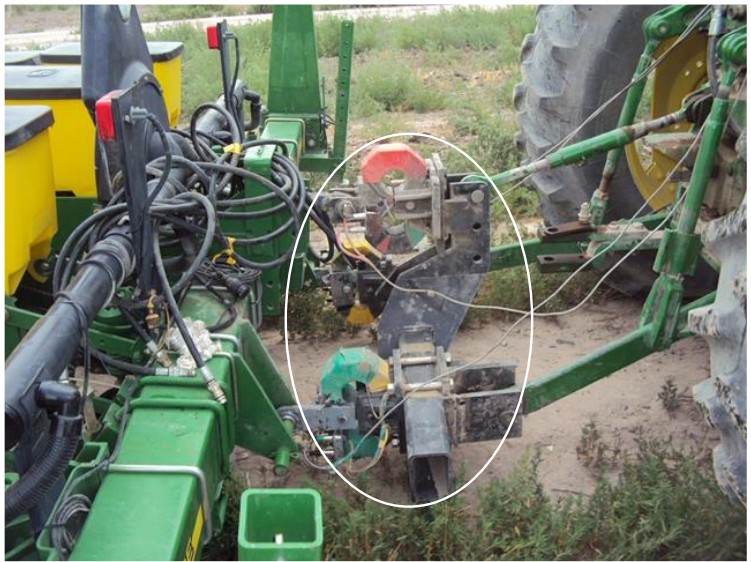

**Figure 1.** An integrated dynamometer for mounted implements (circled) attached to the three-point linkage. The implements are coupled to the dynamometer.

A converter Daq Book 2000 (IOtech, MCC Corporation, Norton, MA, USA) was used to register and transform the signals from the dynamometer at a sampling speed of 20 Hz and pass them to a DBK-43A (IOtech) signal conditioner and store the data in a PC on board in the tractor.

Fuel consumption was measured with a flowmeter S-004 BAICO (BAICO, Guadalajara, Jal, Mexico). This has two sensors; one measures the input flow to the engine and the other measures the return flow to the fuel tank. The sensors send the signals to a data logger Log Book 360 (IOtech). The net consumption is calculated by the difference between the input and return pulses, and this is multiplied by a calibration constant (3.09) to obtain the net volume consumption. The fuel consumption was referenced to the worked area to obtain L ha$^{-1}$.

When performing each tillage operation, working speed (m s$^{-1}$) was measured in three replications per plot, by recording the time that the tractor took to travel 20 m. In the field, working width (m) and working depth (m) were measured using metric tape, stainless steel ruler, and hand level. Average of the three replications for working width, and five replications for working depth, were computed for each plot. Table 3 presents the average values of these variables.

**Table 3.** Average of the data measured for working speed, width, and depth.

| Average of Field Measurements | Disc Plow ARHK-3 | Chisel Plow JD 610 | Disc Harrow RI 2024 | Planter JD Max Emerge |
|---|---|---|---|---|
| Working width (mm) | 800 | 2200 | 2250 | 2700 |
| Working speed (m s$^{-1}$) | 1.11 | 1.10 | 1.24 | 1.27 |
| Working depth (mm) | 237 | 166 | 126.5 | 50 |

### 2.4. Calculation of Energy

The net power (NP) required for each tillage operation was calculated according to Equation (1), by multiplying the net force (kN) to pull the implement, by the working speed (m s$^{-1}$), obtaining

the net power in units of kW; this also can be expressed in units of energy per time (kJ s$^{-1}$), as in Equation (2).

$$NP = F \times V \tag{1}$$

where: NP = net power (kW); NF= net force (kN); V = working speed (ms$^{-1}$).

This can be expressed as:

$$Et = E/t \tag{2}$$

where Et = net energy applied per time during the tillage operation (kJ s$^{-1}$); E= net energy (kN $\times$ m) = kJ; t= time (s).

The calculation of the net energy per hectare (MJ ha$^{-1}$) was made by dividing the energy applied per time for each operation (kJ s$^{-1}$), by the net field work capacity of each tillage operation, as in Equation (4). The net field capacity was calculated, as in Equation (3), from the field data of working speed (km h$^{-1}$) and working width (m) for each tillage operation.

$$NFc = W \times V/10 \tag{3}$$

where NFc = Net field work capacity (ha h$^{-1}$); W = working width (m); V = working speed (km h$^{-1}$).

$$NE = Et/NFc \tag{4}$$

where NE = net energy applied per ha (MJ ha$^{-1}$); Et = energy applied per time during the tillage operation (MJ h$^{-1}$); NFc = net field work capacity (ha h$^{-1}$).

The calculation of the energy applied to the soil mass (J kg$^{-1}$) was computed using Equation (5).

$$Esm = Et/(Sm) \tag{5}$$

where: Esm = energy applied to the soil mass (J kg$^{-1}$); Et = energy applied per time during the tillage operation (J s$^{-1}$); Sm = weight of soil being moved per time during the tillage labor (kg s$^{-1}$).

The energy applied per time is calculated from the net force to pull the implement (N) multiplied by the working speed (ms$^{-1}$).

The weight of the soil being moved per time is calculated as follows:

$$Sm = W \times D \times V \times Bd \tag{6}$$

where: Sm = soil moved per time (kg s$^{-1}$); W= working width (m); D= working depth (m); V = working speed (m s$^{-1}$); Bd= Bulk density of the soil (kg m$^{-3}$).

## 2.5. Calculation of Overall Efficiency

The overall efficiency of the use of fuel by the tillage system that includes the load matching of the tractor and implement, the tractive efficiency, and the engine/power train operating conditions was obtained using Equation (7), by dividing the net energy used for the tillage operation in a hectare and the energy produced by the net volume of fuel used per hectare, similar to that proposed by [35].

$$OE = (NE/(HHVD \times NFC)) \times 100) \tag{7}$$

where OE = Overall efficiency of the use of fuel (%); NE = Net energy applied per ha (MJ ha$^{-1}$); HHVD = Higher heating value of diesel fuel = 38.59 MJ L$^{-1}$; NFC = Net fuel consumption (L ha$^{-1}$).

## 2.6. Bulk Density and Total Porosity

At the end of the summer cropping season 2013, and after three years in 2016, at each tillage treatment, undisturbed core samples were carefully taken in the soil profile from 0 to 20 cm at intervals

of 5 cm. The core sampling was made using cylinders of 50 mm diameter and 50 mm length; samples were processed according to procedures described in [36]. Soil bulk density was calculated as follows:

$$Bd = M/V \tag{8}$$

where Bd = bulk density (gr cm$^{-3}$); M = mass of the dry soil sample (gr); V = volume of sample (cm$^3$).

Total porosity was calculated from the values of bulk density and particle density. The later was determined by the picnometry method [37]. Total porosity was calculated as follows:

$$TP = (1 - Bd/Pd) \times 100 \tag{9}$$

where TP = total porosity (%); Bd = bulk density (gr cm$^{-3}$); Pd = particle density (gr cm$^{-3}$).

## 2.7. Crop Yield

The crop maize was harvested in 2013 and 2014 as fodder. Following the procedure of [38], samples of the crop were cut, weighed as green fodder, and then dehydrated at a temperature of 70 °C for 72 h until depletion of moisture, and so, its weight was constant, to obtain the yield of the dry fodder. This was taken as the dry matter weight of the crop, to calculate yield per hectare.

## 3. Results and Discussion

### 3.1. Draft and Energy for Tillage Operation and Systems

The draft for each tillage implement is presented in Table 4. The draft was calculated with the force measured to pull the implement and the working width. The draft ranged from a minimum of 1.81 kN m$^{-1}$ for the planter to a maximum of 29.89 kN m$^{-1}$ for the disk plow. Working depth was the main factor that influenced the draft. As seen in Table 5, the average working depth was disk plow > chisel plow > disk harrow > planter, so in this order was the implement draft.

**Table 4.** Draft for each implement used in the tillage systems.

| Implement/Year | 2013 | 2016 | 2017 | Average | ASABE Estimate |
|---|---|---|---|---|---|
| | | Draft (kN m$^{-1}$) | | | |
| Disk Plow | 29.89 | 27.99 | 31.10 | 29.66 | - |
| Chisel Plow | 11.28 | 5.32 | 8.16 | 8.25 | 12.60 |
| Disk Harrow | 5.56 | 5.18 | 4.92 | 5.22 | 7.60 |
| Planter | 2.81 | 1.81 | 1.95 | 2.19 | 2.64 |

**Table 5.** Working depth of the implements for each tillage system.

| Tillage Systems | Years | | | Average |
|---|---|---|---|---|
| | 2013 | 2016 | 2017 | |
| | Working Depth (mm) | | | |
| DDP | | | | |
| Disk Plow | 279 | 169 | 263 | 237 |
| Disk Harrow | 100 | 130 | 148 | 126 |
| Planter | 50 | 50 | 50 | 50 |
| CHDP | | | | |
| Chisel Plow | 148 | 119 | 231 | 166 |
| Disk Harrow | 100 | 130 | 150 | 127 |
| Planter | 50 | 50 | 50 | 50 |
| NT | | | | |
| Planter | 50 | 50 | 50 | 50 |

The American Society of Agricultural and Biological Engineers (ASABE) Standard D497.7 [39] defines tillage implement draft as force per unit width. The average measured values of tillage implement draft were compared to those estimated with the equation $D = Fi(A + B(S) + C(S)^2)WT$, where D is the implement draft per unit width; Fi is a dimensionless soil texture adjustment parameter with different values for fine-, medium-, and coarse-textured soils; A, B, and C are machine-specific parameters; S is field speed; W is implement width; and T is tillage depth. The equation of the ASABE Standard overestimates by 20.50% the draft for the planter, 52.66% for the Chisel plow, and 45.60% for the disk harrow. No data for disk plow or similar implement is in the ASABE Standard D497.7. The ASABE coefficients in the equation are for a wide range of soil conditions and consequently cannot be expected to result in accurate estimates for a particular situation. The standard has an expected range of ±35% for the draft for direct planters, so the measured value was within this range, up to ±50% for offset disk harrows, the measured value was also in the range, and up to ±50% for chisel plow, so the measured value is a bit out of the range. In a similar study, the draft for the chisel plow was overestimated by the standard by 69% [40].

The average amount of energy used for each labor, with each tillage system, is observed in Table 6. In the DDP system, primary tillage with the disk plow required the greatest quantity of energy compared to all other implements. For this tillage operation, year to year variations were not significant. For the subsequent tillage operations of harrowing and planting in the conventional system, likewise, there were no significant differences in the amount of energy between years.

In relation to the CHDP system, the primary tillage made with the chisel plow had a significantly lower value of energy in 2016 compared to the other years; this decrease in energy was related to the shallower working depth (Table 6) and the relatively higher moisture content in the soil profile for that year (Table 1). For the secondary tillage operation of harrowing, there were no significant differences between years. Energy for planting in this system was significantly higher in 2013 compared to the other years.

**Table 6.** Specific energy per area used at each labor for the different tillage systems.

| Tillage Systems | Years | | | Average |
|---|---|---|---|---|
| | **2013** | **2016** | **2017** | |
| | MJ ha$^{-1}$ | | | |
| DDP | | | | |
| Disk Plow | 284.59 a | 298.17 a | 311.02 a | 297.93 |
| Disk Harrow | 74.02 a | 50.55 a | 46.56 a | 57.04 |
| Planter | 27.86 a | 25.58a | 20.89 a | 24.78 |
| CHDP | | | | |
| Chisel Plow | 89.02 a | 43.56 b | 74.25 a | 68.94 |
| Disk Harrow | 62.63 a | 40.25 a | 39.60 a | 47.49 |
| Planter | 28.67 a | 13.95 b | 13.09 b | 18.57 |
| NT | | | | |
| Planter | 28.05 ab | 18.59 b | 32.65 a | 26.43 |

Mean values with the same letter in the same row are not significantly different (Tukey, $\alpha \leq 0.05$).

The no-till system had only the labor of direct planting. The lower use of energy in 2016 for this operation coincides with a relatively greater moisture content compared with 2017.

The requirement of energy for the tillage operations depends mainly on the soil resistance, which, in turn is related to moisture content, working depth, and working width. In our study, Disk plow had the highest draft force and energy requirement because of the deeper working depth. Similarly, in other studies with primary tillage using moldboard or disk plows, the demand of drawbar force, and, hence, energy, was higher compared to other implements [41–43].

The quantity of net energy demand, from the tillage implements in conservation systems, was lower than that for the conventional system, commonly used in the region of our study. The results were in general accord with the findings of other studies undertaken in different environmental conditions. [44,45].

In Figure 2, it is observed that the total net energy for each tillage system is significantly different. The energy used by the sequence of operations of the CHDP system saved, on average, up to 64% of energy and NT saved up to 93% of energy, compared with the DDP. These findings are similar to the results obtained by other authors studying conservation tillage [46,47].

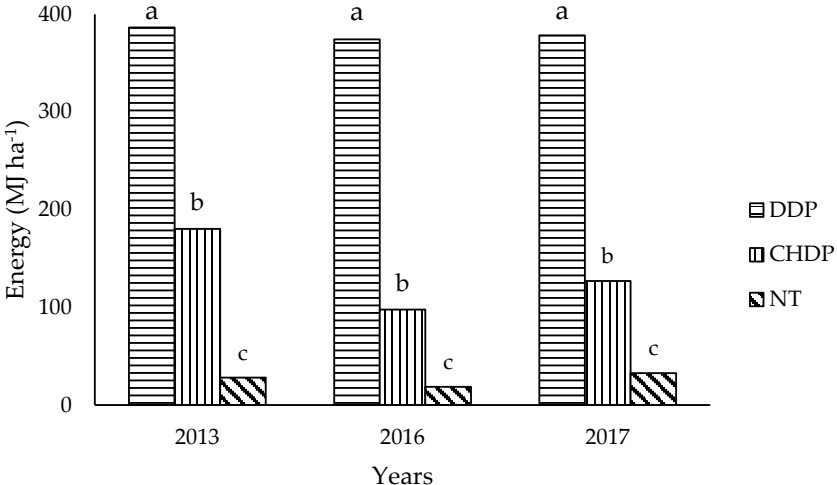

**Figure 2.** Energy per worked area for each tillage system; different letters in columns in the same year have significant differences (Tukey, $\alpha \leq 0.05$).

The amount of energy used is related to the volume of soil worked (Table 7). The DDP system moved an average of 4530 m$^{-3}$ per hectare, while the CHDP system moved 15.5% less and NT system moved 80% less volume of soil, compared to the DDP system. These results show the higher demand for energy for DDP.

**Table 7.** Volume of soil moved for each operation in the tillage systems.

| Tillage Systems/Years | 2013 | 2016 | 2017 | Average |
|---|---|---|---|---|
| | Volume of Soil Moved m$^3$ ha$^{-1}$ | | | |
| DDP | | | | |
| 　Disk Plow | 2790 | 1690 | 2630 | 2370 |
| 　Disk Harrow | 1000 | 1300 | 1480 | 1260 |
| 　Planter | 500 | 1100 | 1100 | 900 |
| CHDP | | | | |
| 　Chisel Plow | 1475 | 1190 | 2310 | 1658 |
| 　Disk Harrow | 1000 | 1300 | 1500 | 1267 |
| 　Planter | 500 | 1100 | 1100 | 900 |
| NT | | | | |
| 　Planter | 500 | 1100 | 1100 | 900 |

*3.2. Specific Draft per Soil Volume and Specific Energy per Moved Soil Mass*

The specific draft force per unit volume of soil disturbed for each implement at each system is shown in Table 8. It can be seen that the highest specific draft per volume is applied with the tools of the planter in contact with soil; this is because the unit of volume of soil disturbed in the planting line is the smallest compared with the other implements, thus resulting in a high energy applied to the worked soil volume.

**Table 8.** Draft applied per volume of soil disturbed for each implement at each tillage system.

| Tillage Systems | Years | | | Average |
| --- | --- | --- | --- | --- |
| | **2013** | **2016** | **2017** | |
| | kN m$^{-3}$ | | | |
| DDP | | | | |
| Disk plow | 115.29 b | 166.89 b | 118.52 b | 133.56 |
| Disk harrow | 66.61 c | 38.88 c | 31.04 c | 45.51 |
| Planter | 457.74 a | 410.17 a | 338.74 a | 402.21 |
| CHDP | | | | |
| Chisel Plow | 92.65 b | 36.50 b | 32.02 b | 53.72 |
| Disk harrow | 55.60 c | 30.96 b | 26.71 b | 37.75 |
| Planter | 470.99 a | 226.73 a | 226.23 a | 307.81 |
| NT | | | | |
| Planter | 460.82 | 244.31 | 384.35 | 363.16 |

Mean values with the same letter in a column in the same tillage system are not significant different (Tukey, $\alpha \leq 0.05$).

As the volume of worked soil increases the specific draft per volume diminish. Implements with the bigger working width, such as the chisel plow and disk harrow, applied less specific draft per volume to the soil compared to the disk plow that have less width. Similar results in tillage intensity are presented in [48], where a moldboard plow working at greater depth and less width applied a greater specific draft energy to the soil, compared to a field cultivator working at greater width and less depth.

Figure 3 presents the total specify energy applied to the soil mass by the sum of operations in each tillage system. The sum of tillage operations of DDP was the system which applied the greatest amount of energy to the soil mass to achieve the seedbed preparation. It can be noticed that NT, with only one operation of planting, exerts a great deal of energy (a high loading rate) to a small soil mass unit; it represents from 40% to 80% of that exerted for all the operations of the DDP in the different years studied. Specific energy applied to the soil mass is required to reduce the size of soil aggregates. The amount of energy depends on the stress loading rates [49]. In the NT system, it is required that the tools of the planter, in contact with the soil, leave a condition of small aggregates for an adequate soil seed contact. These tools are designed to achieve this condition with only one pass.

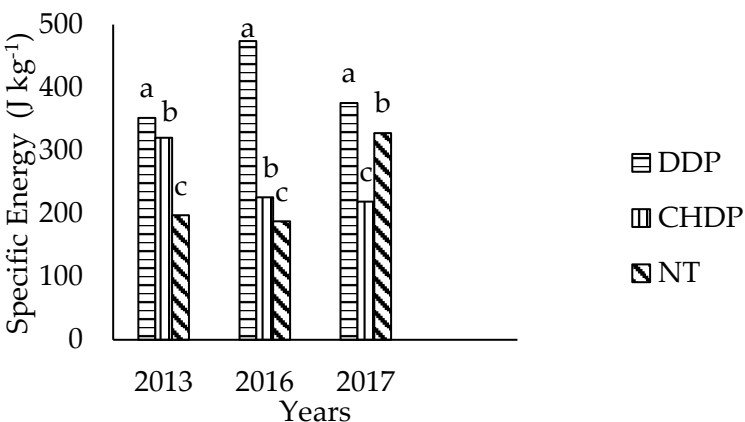

**Figure 3.** Specific energy applied to the soil mass for each tillage energy; different letters in columns in the same year are significantly different (Tukey, $\alpha \leq 0.05$).

### 3.3. Fuel Used by Worked Area

Fuel consumption was registered only for the years 2016 and 2017.

In Table 9, the fuel consumption for each operation in the tillage systems can be seen. Disk plowing recorded the highest fuel consumption per worked area while showing the lowest fuel consumption for the planting operation.

**Table 9.** Fuel consumption per worked area for each operation in the tillage systems.

| Tillage Systems | Years | | Average |
|---|---|---|---|
| | 2016 | 2017 | |
| | L ha$^{-1}$ | | |
| DDP | | | |
| Disk plow | 25.35 a | 31.36 a | 28.35 |
| Disk harrow | 14.88 a | 12.93 a | 13.90 |
| Planter | 13.40 a | 10.90 a | 12.15 |
| CHDP | | | |
| Chisel plow | 23.73 a | 20.68 a | 22.20 |
| Disk harrow | 15.78 a | 14.00 a | 14.89 |
| Planter | 9.26 a | 9.64 a | 9.45 |
| NT | | | |
| Planter | 11.87 a | 10.27 a | 11.07 |

Mean values with the same letter in the same row are not significantly different (Tukey, $\alpha \leq 0.05$).

The sequence of operations of DDP had the greatest fuel consumption. This is similar to other studies where the highest fuel consumption was logged by a conventional system of deep plowing, compared to a reduced tillage system, which required up to 58% less fuel [50]. Increase in fuel consumption is generally related to increase in working depth. Consequently, disk plowing had the greatest fuel consumption, followed by chisel plowing, disk harrowing, and planting.

Figure 4, shows the fuel consumption per area for each tillage system. In the year 2016, fuel consumption of DDP and CHDP were not significantly different. For the two years, on average, the CHDP system used 14.45% less fuel and NT 79.65% less fuel compared to the DDP. These results agree with the findings of [51] where the establishment of a wheat crop required up to seven times more volume of fuel compared with direct planting.

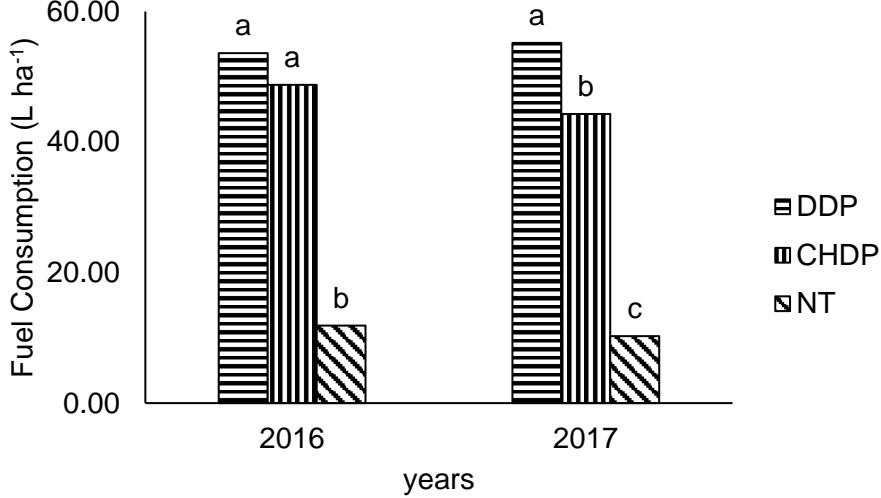

**Figure 4.** Fuel consumption per area for each tillage system; different letters in columns in the same year are significantly different (Tukey, $\alpha \leq 0.05$).

### 3.4. Overall Efficiency in the Use of Energy from the Fuel

Figure 5 presents the percentage of the overall efficiency in the use of energy from the diesel fuel. For this study we used 38.597 MJ L$^{-1}$, which is the higher heating value of a liter of diesel according to The American Society for Testing and Materials (ASTM) D975 [52].

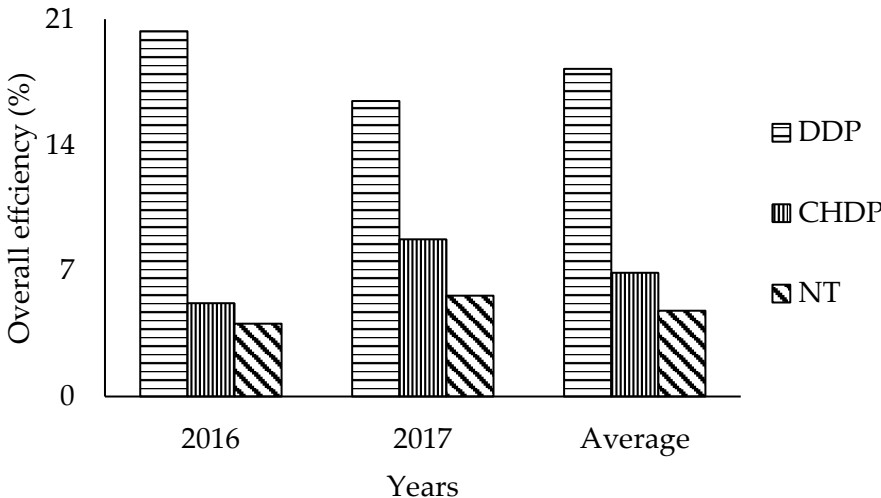

**Figure 5.** Overall efficiency in the use of energy from fuel by the tillage systems.

The overall efficiency in the use of energy from the fuel was calculated by dividing the net energy used per area by the total energy available in the volume of fuel used per area. This is calculated for each tillage system. The overall energy efficiency includes the load matching of the tractor and implement. A tractor–implement combination having an overall energy efficiency below 10% indicates poor load matching or/and low tractive efficiency, while a value above 20% indicates a good load match or/and high tractive efficiency [53]

Average efficiency in the use of energy by the tillage systems was quite low. For DDP resulted in 18.23%, for CHDP 6.88%, and 4.77% for NT. Similar results were found in other studies; the overall efficiency of energy use was in a range from 11% to 20.08% for different tillage implements [54,55].

Considering the total energy available in the fuel that was consumed per hectare; the overall efficiency indicates the percentage that is being used as net energy per hectare in the sequence of tillage operations for each system. Energy that is not used for the tillage operations is used to move the tractor, other losses are by heat, friction, transmission, and slippage, etc.

The low efficiency in the use of energy available from the fuel indicates an inadequate combination of tractor–implement for CHDP and NT. The equipment used was typical of the production methods used in our area, where the farmers normally only own one tractor, which they use for all their operations. The rated power is based on the demand of the disc plowing, which is the heaviest operation. For this reason, the power of the tractors used for the local conventional tillage is underutilized, because of the relatively light tillage operations of CHDP and NT. So, account should be taken of all the factors involved to increase, in the future, the efficiency in the use of energy in the tillage operations that demand less power. This could be achieved using lighter tractors, increasing the working width by using bigger implements, or increasing the working speed.

From the results of our study, conservation tillage (CHDP and NT) reduce the use of total energy for soil preparation compared to DDP significantly. But it is important that the amount of energy that corresponds to each system and operation is used with efficiency. According to the characteristics of the tractors in the region, attention has to be paid to a good match between power source–implement to achieve a good load matching and tractive efficiency and increase the overall efficiency.

### 3.5. Effect of Tillage on Bulk Density and Porosity of the Soil

The use and cost of energy in the tillage systems is important, but a positive effect on soil properties for good crop development is the final goal of tillage. As an indicator of this effect, soil bulk density (Bd) was measured and soil porosity (Sp) calculated at the end of the summer cropping season in the tillage treatments in 2013 and after three years in 2016. This was done to look at the effects in the short term of the tillage systems on these soil physical variables. In Figure 6, it can be observed that the bulk density in 2013 was significantly lower only in the surface layer in the CHDP treatment compared to NT and DDP. For the deeper layers, there were no significant differences between treatments.

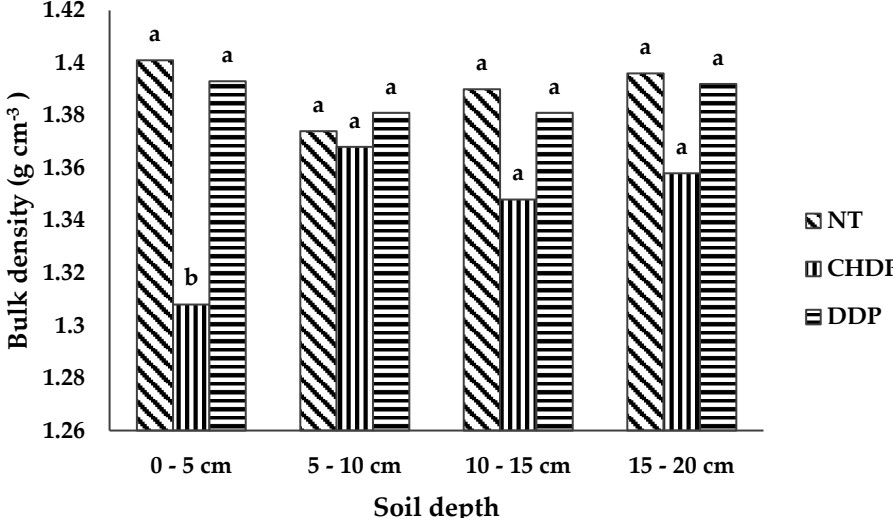

**Figure 6.** Bulk density in the soil profile at intervals of 5 cm at the beginning of the experiment in 2013. Mean values with the same letter in the same depth interval are not significantly different (Tukey, $\alpha \leq 0.05$).

Total porosity depends on the value of Bd; it can be observed in Figure 7 that at the beginning of the experiment, total porosity was very similar for the treatments. The exception was the porosity in CHDP at the top layer that was significantly higher than DDP and NT.

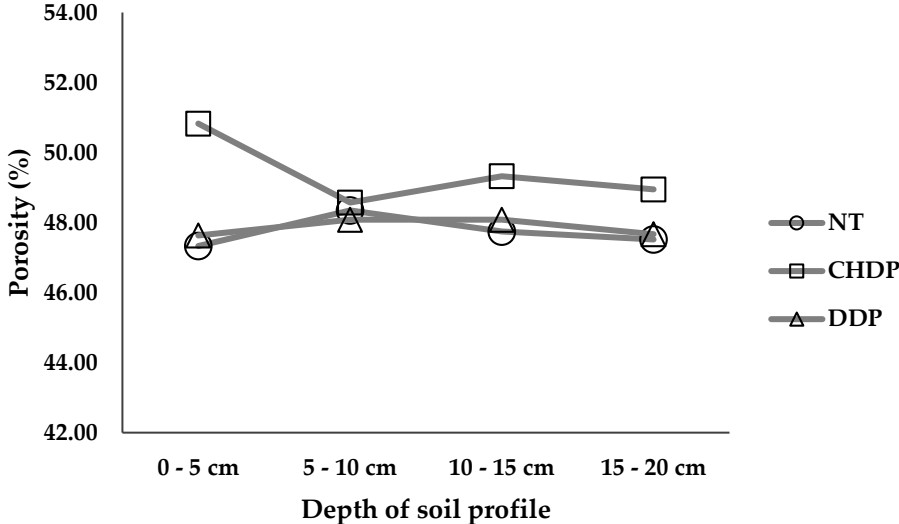

**Figure 7.** Total porosity in the soil profile at intervals of 5 cm at the beginning of the experiment in 2013.

After three years, the Bd decreased for the NT at all depth intervals and increased for the CHDP. The values of the Bd in the DDP decreased in the surface layer and the bottom layer and increased in

the middle layers compared with the values of 2013 (Figure 8). The values for NT at all depths are significantly lower compared to CHDP and DDP.

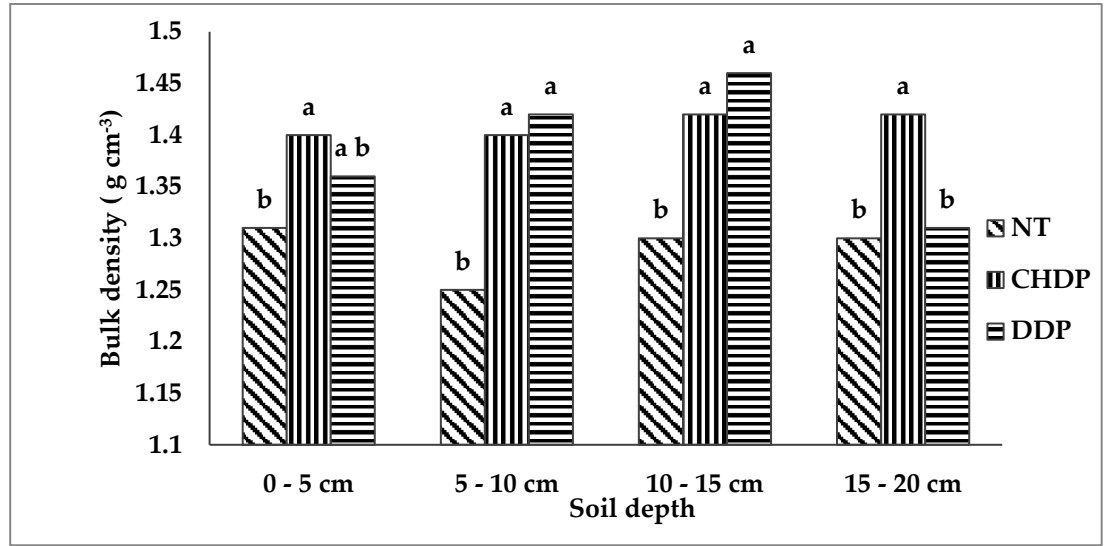

**Figure 8.** Bulk density in the soil profile at intervals of 5 cm after three years of the tillage treatments in 2016. Means values with the same letter in the same depth interval are not significantly different (Tukey, $\alpha \leq 0.05$).

Bulk density is an indicator of soil quality, and is often used for evaluating tillage effects [56]. If the density decreases (Figure 8), then the porosity increases (Figure 9), and the soil has more capacity for infiltration and storage of water, thus, improving its availability for the plants. [57]. After three years, the no-till treatment had the lowest bulk density and provided the highest total porosity (Figure 9).

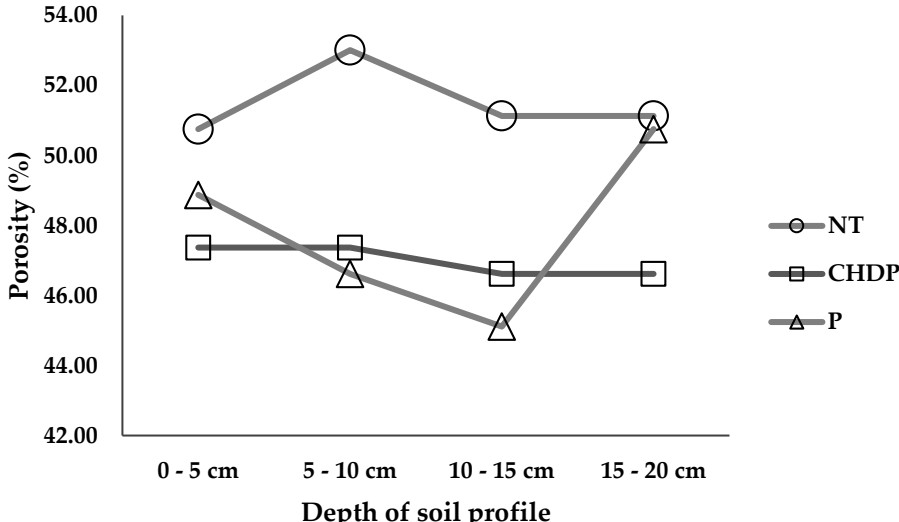

**Figure 9.** Soil porosity at intervals of 5 cm after three years of the tillage treatments in 2016.

In similar studies, the results of the changes in the soil bulk density are contradictory. There are experiments in which there are higher increases with time of bulk density values under conventional compared with conservation tillage [58]. But other studies report that the no-till treatment had the highest bulk density and provided the lowest total porosity [59].

Other result shows that in the short term (after two years) direct planting treatment in clay soil in humid climate had a better physical condition and root distribution index than conventional tillage [60].

Another study, in a humid subtropical area of Mexico, showed that bulk density was significantly higher after three years with no tillage compared with conventional tillage [61].

Contrasting effects of soil management experiments in bulk density are common. Effects are related to the management of the machinery (weight of implements, number of passes) and to the soil water content at the moment of performing the tillage operation. [62]. From the point of machinery management, it is possible that in our experiment, the systems that have more tillage operations (DDP and CHDP), increased the soil bulk density compared to NT that has only one operation.

### 3.6. Effect of Tillage on the Dry Matter Yield of Maize

The reduction of tillage operations and, thus, the cost of energy should not affect the yields of crops. For the adoption of conservation tillage, the farmer must have a good balance in the cost–benefit of the production system. To study the effect of tillage systems on the dry matter, the yield of maize was measured for the seasons 2013 and 2014. There were no significant differences between treatments as shown in Figure 10. However, in both years the yield of NT was 16.2% and 12.8% higher than DDP. The yield CHDP was 10.8% and 11.5% higher than DDP.

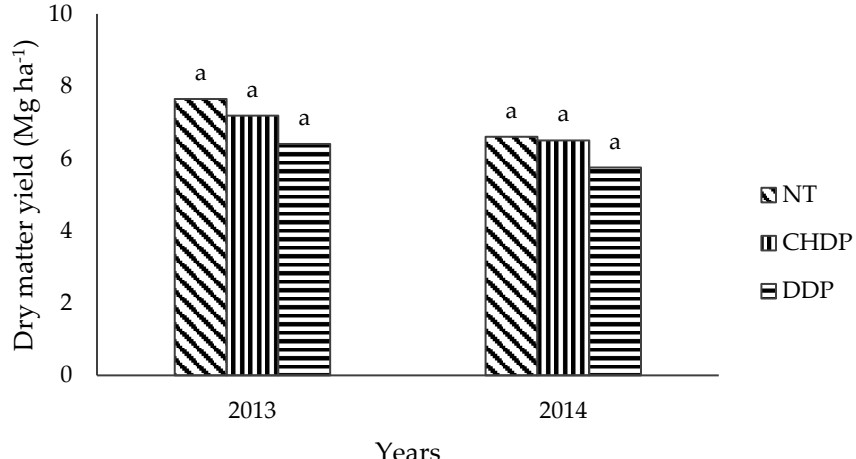

**Figure 10.** Dry matter yield of maize for the three tillage systems in the seasons 2013 and 2014. Mean values with the same letter in the same year are not significantly different (Tukey, $\alpha \leq 0.05$).

Our study agrees with other similar studies [63] where maize yield was higher with conservation tillage practices in an area of limited rainfall. This might be attributed to moisture conservation in low rainfall areas under conservation agriculture.

No-till practices enhance water use efficiency when residues are retained, this is why conservation tillage performs better than conventional tillage under limited water conditions [10]. Conservation tillage can increase crop productivity in dry climates. Hence, it may become an important climate-change adaptation strategy for these regions of the world. [64,65].

### 4. Conclusions

The conservation tillage systems CHDP and NT demand less specific energy per area compared to the conventional system DDP. It is possible to save 64% in energy with CHDP and 93% in energy with P compared with the energy expenditure of DDP.

For each of the tillage systems, the planting operation applied the highest specific energy to the soil mass. This is because it moves less volume of soil.

The sequence of tillage operations in the DDP system had the highest fuel consumption per worked area compared to the other systems. On average, the CHDP system can save 14.45% fuel, and the NT can save 79.65% of fuel, compared with the DDP system.

The overall efficiency in the use of energy was better in DDP compared to CHDP and NT. Attention has to be paid to a good match between tractor–implement to achieve a good load matching and tractive efficiency in conservation systems.

The NT system significantly decreases the values of the soil bulk density after three years, compared to DDP and CHDP. P increases the total porosity, leaving the soil in better condition for intake and storage of water.

NT and CHDP had a higher yield than DDP in dry matter yield of maize, although not of statistical significance.

In the semiarid areas of Mexico, NT and CHDP are a good option to decrease the use of energy for soil preparation. NT also improved the physical condition of the soil.

**Author Contributions:** Conceptualization, M.C.-Z. and S.C.-M.; Methodology, M.C.-Z., S.C.-M., A.Z.-G., A.L.-V.; Software, A.L.-V.; Validation, A.Z.-G., A.L.-V. and M.C.-Z.; Formal Analysis, M.C.-Z.; Investigation, A.L.-V., M.C.-Z., S.C.-M.; Resources, A.Z.-G., M.M.-D.; Data Curation, M.C.-Z., A.L.-V.; Writing-Original Draft Preparation, M.C.-Z., A.L.-V.; Writing-Review and Editing, M.C.-Z., M.M.-D.

**Funding:** This research received no external funding, internal funds were provided by Universidad Autonoma Agraria Antonio Narro (UAAAN).

**Acknowledgments:** The authors acknowledge the technical support from the staff of Agricultural Machinery Department of Universidad Autonoma Agraria Antonio Narro and to David Mann for the revision of the manuscript.

**Conflicts of Interest:** The authors declare no conflict of interest. The internal funder UAAAN had no role in the design of the study; in the collection, analyses, or interpretation of data; in the writing of the manuscript, and in the decision to publish the results.

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
