# Peer review of "Comparison of Energy Used and Effects on Bulk Density and Yield by Tillage Systems in a Semiarid Condition of Mexico"

_agronomy, doi:10.3390/agronomy9040189_

Reviewer 1 Report

From the reviewer point of view all of the remarks were reflected correctly. 

Author Response

Response to Reviewer 1 Comments

Point 1

Comments and Suggestions for Authors

From the reviewer point of view all of the remarks were reflected correctly. 

Response 1

Thank you very much for the time deserved to review our manuscript, your suggestions, comments and observations were very helpful to improve the manuscript. We are very grateful for your professional work throughout the revision process

Reviewer 2 Report

The authors report on a series of measurements of tillage draft, fuel use and crop response in a clay loam soil.  While there is some value in the work, there are several problems with the system comparisons:

Vertical tillage as described is not consistent with current use of the term.  Vertical tillage generally refers to tillage with coulters (not concave disks) that work the soil typically no deeper than 10 cm.  What was describe here would be named a chisel/disk/plant system versus a disk plow/disk/plant system.

The specific draft measurements could be useful if presented as pulling force per unit of depth, pulling force per unit volume of soil disturbed, etc., however that data is obscured by presenting it in various forms of fuel efficiency.

I have concerns about the measurements of fuel use efficiency because the same tractor was used in all three comparisons.  Typically, well-matched tractor/implement combinations would be used for such comparisons with the tractor sized for the highest draft operation.  Using the tractor sized (I assume) for the disk plow and comparing it to the fuel efficiency for the chisel plow and no-till system presents data biased against the lower draft systems.  You can also see that the rear tires are quite worn which would also affect fuel use efficiency.  2-WD or MFWD?  Front to rear weight distribution?

Information is lacking on the implement specification.  Concavity of the disk plow disks and the disk used (tandem disk?)  Disk spacing?  Configuration of the planter units and type/number of coulters/openers used.

How do the draft measurements compare to ASABE standards for similar implements?

The implement draft/soils/crop response information has to most potential value of this work and may help convince farmers and advisors of the potential savings possible by putting well-matched tractors and implements in the field.

Author Response

Response to Reviewer 2 Comments

Thank you very much for the time deserved to review our manuscript, your suggestions, comments and observations were very helpful to improve the manuscript.

Point 1

Comments and Suggestions for Authors

The authors report on a series of measurements of tillage draft, fuel use and crop response in a clay loam soil.  While there is some value in the work, there are several problems with the system comparisons:

Vertical tillage as described is not consistent with current use of the term.  Vertical tillage generally refers to tillage with coulters (not concave disks) that work the soil typically no deeper than 10 cm.  What was describe here would be named a chisel/disk/plant system versus a disk plow/disk/plant system. 

Response 1

We think there are not a standard nomenclature yet for vertical tillage, some authors (Kovach et al 2012) name a vertical tillage implement with shallow concave disk blades, they mention that the shallow concave disk blades have a concavity of between approximately 1.25 and 1.69 inches over an outer diameter of approximately 20 inches. In other paper, (Reicosky 2015) it is mentioned a vertical tillage implement with disks. Other authors (Campos et al, 2015) name a vertical tillage tool a chisel type implement.

With respect to our manuscript in order to be clear about the tillage systems used in this study, we think it is better as you suggest to name the systems according to the tillage operations that we used:  Disk Plow/Disk/Planter (DDP), for the tillage most used in our region and for the alternative systems that we used, chisel/disk/planter (CHDP), and the No-Tillage (NT) that is direct planting. According to this, we changed the nomenclature of the tillage systems (CT, VT) used throughout the manuscript.

Point 2

The specific draft measurements could be useful if presented as pulling force per unit of depth, pulling force per unit volume of soil disturbed, etc., however that data is obscured by presenting it in various forms of fuel efficiency.

Response 2

We agree with your observation, that it is useful to present information on specific draft. We add information on draft for each tillage implement in subchapter 3.1 from line 256 to 260 and table 4. Also information on draft per unit volume of soil disturbed in subchapter 3.2 we add information of specific draft per volume of soil disturbed in Lines 315 to 322 and table 8

Point 3

I have concerns about the measurements of fuel use efficiency because the same tractor was used in all three comparisons.  Typically, well-matched tractor/implement combinations would be used for such comparisons with the tractor sized for the highest draft operation.  Using the tractor sized (I assume) for the disk plow and comparing it to the fuel efficiency for the chisel plow and no-till system presents data biased against the lower draft systems. 

Response 3

The reason of doing the comparison of tillage operations with the same tractor is stated in the manuscript (lines 379 to 386). The farmers normally only own one tractor, which they use for all their operations. If they change the tillage operations or even the system, they still using the same tractor so, we have the opportunity of quantify the efficiency in order to be aware in the recommendations that we have to make and take in to account all the factors involved in order to increase in the future, the efficiency in use of energy in the tillage operations that demand less power. This could be achieved by different alternatives using lighter tractors, increasing the working width using bigger implements, or increasing the working speed.

Point 4

You can also see that the rear tires are quite worn which would also affect fuel use efficiency.  2-WD or MFWD?  Front to rear weight distribution?

Response Point 4

Yes, worn tires affect the fuel use efficiency we did not quantify that in this study. In our future work about increase the efficiency of use of energy it must be take into account.

The tractor is 2-WD with a Front to Rear weight distribution of 35%, 65%. This information has been added in materials and methods in Line 133 - 134

Point 5

Information is lacking on the implement specification.  Concavity of the disk plow disks and the disk used (tandem disk?)  Disk spacing?  Configuration of the planter units and type/number of coulters/openers used.

Response 5

The information on concavity of the disk (Plow and Harrow) disk spacing, type of harrow (Offset Disk harrow), configuration of the planter, type/number of openers has been added in Table 2 (line 124) in the characteristics of each implement

Point 6

How do the draft measurements compare to ASABE standards for similar implements?

Response 6

The average draft measurements per year and for the three years are shown now in table 4 (line 263). Estimates of the draft were made according with the Standard ASABE D497.7 MAR2011 (R2015). From line 264 to line 276 it is discussed the results comparing the values measured with the estimates. The draft is overestimated for the implements used (no coefficients in the standard are for disc plow), but only the chisel plow is a bit out of the expected range.

Point 7

The implement draft/soils/crop response information has to most potential value of this work and may help convince farmers and advisors of the potential savings possible by putting well-matched tractors and implements in the field. 

Response 7

Yes, we believe that conservation tillage systems can have good results in our region. Not only saving energy but improving the soil condition and maintaining yields. But when we change to conservation tillage it is necessary to fine tune the matching of the implement-power source. We will continue in this line to get further information to provide farmers and policymakers to get adequate decisions.

Reviewer 3 Report

The paper "Comparison of Energy Used by Tillage Systems in a Semiarid Condition of Mexico" focusses mainly on the comparison between different tillage systems of energy use.

I think that many aspects are not well reported and therefore I  suggest a major revision.

Introduction should be greatly improved: improvement of paragraphs organization is required. For instance, L56-L60 mixes economic and environmental factors. In addition, similar concepts have been reported in previous sentences, e.g. L47-L49. Authors should better organize the introduction.

A geographical context in introduction is lacking, that justifies the importance of your research.
Moreover there is not information on why different climatic conditions should show different results from those already reported in the literature.

Some tenses shoul be reviewed. E.g. L34 and L35 (could and will?).

L43: You should be careful on reporting only benefits from NT, as it may cause also negative effects on soil and therefore yield, depending on soil, climate, management.

Material and methods: How long was the experiment? How was soil moisture measures, which intervals (L85)? Evaporation or evapotranspiration (L83)? Style should be greatly improved, e.g. sentences without subject (L89), punctuation (e.g. L94). Experimental design is confusing: L96, what do you mean? That Experimental design is fully randomized?  2.4: ParagraphThe concept of overall efficiency is a bit confused and I did not understand what does it represent.

I think results should focus more on the effects of tillage on soil properties and yields, while too many factors are related to energy use. Also, I do not understand the point of calculating the energy use per soil mass. My impression is that the second part of results was added (yields, soil properties) to the paper, but  a real integration with the first part (energy use) is lacking.

Results on bulk density and porosity, and yield, are surprising me. How do you explain higher production and porosity in NT compared to VT? Information on it are lacking.

Author Response

Response to Reviewer 3 Comments

We are grateful for the time deserved to review our manuscript, your suggestions, comments and observations were very helpful to improve the manuscript.

Point 1

Comments and Suggestions for Authors

The paper "Comparison of Energy Used by Tillage Systems in a Semiarid Condition of Mexico" focusses mainly on the comparison between different tillage systems of energy use.

I think that many aspects are not well reported and therefore I suggest a major revision.

Introduction should be greatly improved: improvement of paragraphs organization is required. For instance, L56-L60 mixes economic and environmental factors. In addition, similar concepts have been reported in previous sentences, e.g. L47-L49. Authors should better organize the introduction.

Response 1

According to your observations in the introduction and for throughout we improve the writing with the help of a colleague native English speaker trying to convey the information connecting the paragraphs in a better organization following a clearer line of thought.

As indicated in your observation, we arranged the economic issues (cost of energy, prices and production cost) from line 46 to line 53. Then made a transition between the concepts treated (L 54- 55) and continue with environmental and conservation related issues. Improvements based in all the reviewers observations are highlighted in green.

Point 2

A geographical context in introduction is lacking, that justifies the importance of your research.Moreover there is not information on why different climatic conditions should show different results from those already reported in the literature.

Response 2

From L 71 to 75 we add a geographical context and the importance of the region in agricultural production and the main problems faced and the need of technical information. Lines from 76 to 84 give information about contrasting results reported in the literature of tillage systems in different conditions. So it is important to investigate the results of tillage in particular conditions as is the objective of this work.

Point 3

Some tenses should be reviewed. E.g. L34 and L35 (could and will?).

Response 3

This has been reviewed and corrected highlighted in green (lines 36 and 37). A colleague English native speaker has reviewed to improve the writing of all the manuscript.

Point 4

L43: You should be careful on reporting only benefits from NT, as it may cause also negative effects on soil and therefore yield, depending on soil, climate, management.

Response 4

Information in this sense has been put now in lines 44-45  

Point 5 and Responses 5

Material and methods: How long was the experiment?

In line 97 it was added the seasons that the experiment was conducted.

How was soil moisture measures, which intervals (L85)?

Information about the soil moisture sampling and calculation is now in line 103-105

Evaporation or evapotranspiration (L83)?

The term evaporation (now line 101) is correct; the data was measured in a standard evaporation pan at the meteorological station.

Style should be greatly improved, e.g. sentences without subject (L89), punctuation (e.g. L94).

This has been attended, sentences and paragraphs now in lines 109 to 117 has been improved and revised 

Experimental design is confusing: L96, what do you mean? That Experimental design is fully randomized? 

Corrected in Line 119 the experimental design is fully randomized

2.4: ParagraphThe concept of overall efficiency is a bit confused and I did not understand what does it represent. 

Now the subchapter is 2.5 and we added information in lines 217-218 to explain more the concept of overall efficiency

Point 6

I think results should focus more on the effects of tillage on soil properties and yields, while too many factors are related to energy use. My impression is that the second part of results was added (yields, soil properties) to the paper, but a real integration with the first part (energy use) is lacking.

Respond 6

According to your recommendation we made a better integration in the manuscript of the information on soil properties and yield. We added some information from the inclusion in the title (Line 2 -3) throughout all the manuscript: abstract (20-21, 25-26), introduction (lines 37-38, 76-84, 88-89 92) materials and methods (subchapters 2.6 starting in line 228 and subchapter 2.7 starting in line 249), results and discussion, (subchapters 3.5 starting in line 397 and subchapter 3.6 starting in line 459) and conclusions).

Point 7

Also, I do not understand the point of calculating the energy use per soil mass.

Respond 7

This is mentioned in lines 329 to 337. The calculation of energy use per soil mass is important because normally in Mexico we have to add three or more operations (DDP) that disturbed a high volume of soil to achieve the condition required for the seedbed. In this case with one tillage operation (P) direct planting it is applied a high amount of energy per volume and mass of soil to achieve the same result of tillage. This is because the tools of the planter work a very low volume of soil in the planting line.

Point 8

Results on bulk density and porosity, and yield, are surprising me. How do you explain higher production and porosity in NT compared to VT? Information on it are lacking. 

Respond 8

The only explanation of the decrease of bulk density thus the increase in porosity is the reduction of machinery traffic in Direct planting compared to the other systems. In lines 452 to 457 there is information in this point.

In relation to the yield, the differences in dry matter yield among systems were no significant. We have no further information at the moment. The only explanation that we have for the small differences is that explained in lines 473 to 479, in the conservation tillage systems NT and CHDP the dry matter yield was a bit higher probably because the surface residues avoid evaporation conserving moisture in the soil profile. As other studies indicate yield are higher due to more availability of moisture.

 Round  2

Reviewer 3 Report

I think that th uuthors did a good job with the revisions and all all of the remarks were addressed correctly.

This manuscript is a resubmission of an earlier submission. The following is a list of the peer review reports and author responses from that submission.

Round  1

Reviewer 1 Report

On the basis of the studied, I note that the authors have met my requirements. This article is focused on the comparison of energy used by soil cultivation systems in semiarid conditions in Mexico. The authors applied three ways of soil cultivation in 2013, 2016 and 2017, and the following conventional soil cultivation (CT), vertical machining (VT) and no machining (NT). The variables of drawbar force, working speed, width and depth, fuel consumption were measured under an experimental scheme of random blocks. The authors focused on bulk density in soil, the results were added to the post. Based on the required contribution, the calculations show the energy and performance calculations, overall efficiency, bulk density and total porosity. The charts and values of the results were added. The results and discussion have been expanded on the effect of tillage in two physical characteristics of the soil. The conclusion has been expanded and linked to agronomy side.

Overall reassessment:

The reviewer point of view all of the remarks were reflected correctly.

Other comments:

Unify the label in relation to: : d/t = Working speed (ms-1) or v = working speed (km h-1) or V= working speed (m s-1)

Reviewer 2 Report

I do not think this paper is suitable for publication in its present form. THe results presented would be interesting in an LCA of global warming potential of different agricultural practises for example, where accurate measurments of energy consuption during Field operations would be very useful. however, the results do not stand up very well on their own. Tee conclusions that energy consumption is higher in conventional till than no till is hardly surprising. Effects on soils are also considered, but not yield, which I think would be interesting, also a now difference result.

In short, i would urge the authors to rather publish these data as part of larger LCA study.